# U-Net-Based Deep Learning Hybrid Model: Research and Evaluation for Precise Prediction of Spinal Bone Density on Abdominal Radiographs

**DOI:** 10.3390/bioengineering12040385

**Published:** 2025-04-03

**Authors:** Lixiao Zhou, Thongphi Nguyen, Sunghoon Choi, Jonghun Yoon

**Affiliations:** 1Department of Mechanical Design Engineering, Hanyang University, 222, Wangsimni-ro, Seongdong-gu, Seoul 04763, Republic of Korea; zhoulixiao39@gmail.com (L.Z.); npthong2511@gmail.com (T.N.); 2Department of Mechanical Engineering, BK21 FOUR ERICA-ACE Center, Hanyang University, Ansan 15588, Republic of Korea; 3Department of Orthopedic Surgery, College of Medicine, Hanyang University, Seoul 04763, Republic of Korea; spineshchoi@hanyang.ac.kr; 4Department of Mechanical Engineering, Hanyang University, Ansan 15588, Republic of Korea; 5AIDICOME Inc., Ansan 15588, Republic of Korea

**Keywords:** bone mineral density, osteoporosis, U-Net model, artificial neural networks, spinal X-rays

## Abstract

Osteoporosis is a metabolic bone disorder characterized by the progressive loss of bone mass, which significantly increases the risk of fractures. While dual-energy X-ray absorptiometry is the standard technique for assessing bone mineral density, its use is limited in high-risk female populations. Additionally, quantitative computed tomography offers three-dimensional evaluations of bone mineral density but is costly and prone to motion artifacts. To overcome these limitations, this study proposes a hybrid model integrating U-Net and artificial neural networks, specifically focusing on abdominal X-ray images in the anteroposterior view for detailed skeletal analysis and improved accuracy in L2 vertebra mineral density measurement. The model targets female patients, who are at a higher risk for spinal disorders and osteoporosis. The U-Net model is employed for image preprocessing to reduce background noise and enhance bone tissue features, followed by analysis with the artificial neural network model to predict bone mineral density through nonlinear regression. The performance of the model, demonstrated by a high correlation coefficient of 0.77 and a low mean absolute error of 0.08 g per square centimeter, highlights its significance and effectiveness, particularly in comparison to dual-energy X-ray absorptiometry.

## 1. Introduction

Osteoporosis, a common metabolic bone disorder, is characterized by the progressive loss of bone mass density. Healthy bones exhibit a honeycomb-like structure with balanced voids and spaces, which is critical for maintaining proper bone density. Osteoporosis leads to a systemic decline in bone mass, strength, and microarchitecture, significantly increasing the risk of fragility fractures [1]. In 2020, an estimated 12.3 million individuals in the United States over the age of 50 were expected to have osteoporosis [2]. The lifetime fracture risk in patients with osteoporosis can be as high as 40%, with fractures most frequently occurring in the spine, hip, or wrist [3,4,5]. Notably, one in three women over the age of 50 years experiences an osteoporosis-related fracture [6]. Such fractures often necessitate hospitalization and carry an elevated risk of further complications, such as pneumonia or thromboembolic disease, owing to prolonged immobilization [7]. A considerable body of research underscores the necessity of early intervention for those at high risk of fractures, with evidence indicating a marked reduction in fracture incidence following prompt diagnosis [8,9,10]. Consequently, early detection and preventive measures for osteoporosis are vital for enhancing the quality of life of the elderly and reducing associated healthcare burdens.

To mitigate the influence of osteoporosis and prevent spinal fractures, the regular monitoring of bone mineral density (BMD) is essential as part of early diagnosis and intervention strategies. A low BMD is a key predictor of osteoporotic fractures, particularly those of the spine. Particularly for L2 BMD detection, studies have shown that failure characteristics, such as load, stiffness, and energy to failure, are significantly correlated with the BMD value at L2 among L2-L4 [11]. Dual-energy X-ray absorptiometry (DXA), the most widely used technique for measuring BMD, uses low-dose X-ray beams to assess bone density by comparing the absorption of two energy levels, making it effective for evaluating overall bone health [12]. However, DXA screening is underutilized among women at increased fracture risk; nearly half of the female Medicare beneficiaries in the United States do not undergo screening [13], and screening rates among high-risk groups in the United States are as low as 6.5% [14].

In contrast, quantitative computed tomography (QCT) offers a three-dimensional assessment of BMD and provides detailed insights into bone quality, particularly of the spine and hips [15]. However, QCT is limited by its high cost, and the imaging process requires several minutes because of the use of low-power X-ray tubes, which increase potential for motion artefacts [16]. Considering the limitations of these existing technologies, there is an apparent need for a more cost-effective and efficient method for screening osteoporosis.

## 2. Recent Research on Spine BMD Prediction

Given the limitations of current BMD measurement techniques, developing a method that is both cost-effective and efficient is essential, especially for the large-scale screening and early detection of osteoporosis in high-risk populations. Recent advancements in medical image analysis have increasingly leveraged machine learning and deep learning algorithms. Several studies have explored the use of X-ray imaging to predict the BMD, offering innovative alternatives to traditional methods.

Rachidi et al. [17] applied innovative Laws’ masks to analyze trabecular bone structural changes in X-ray images to predict osteoporosis. This method extracts textural features from trabecular bone structures using Laws’ masks, which capture texture properties like edges and ripples. The masks filter bone images to generate texture energy maps, from which meaningful features are extracted for osteoporosis classification. While innovative, this approach is limited by its reliance on a single type of feature and its lower sensitivity compared to traditional BMD measurements. The use of Laws’ masks is particularly effective for textures with a short coherence length or correlation distance, making it suitable for analyzing the fine and complex structure of trabecular bone. However, the method’s performance may be affected by factors such as the image quality and the presence of noise. Future work could explore the integration of Laws’ masks with other feature extraction techniques to improve the accuracy and robustness of osteoporosis prediction.

Zhang et al. [18] utilized deep convolutional neural networks (CNNs) to screen for osteoporosis and osteopenia in lumbar X-ray images, using DXA-derived BMD values as the reference standard. Their approach involved training a CNN architecture on a large dataset of lumbar X-ray images to automatically extract hierarchical features indicative of bone density variations. The CNN model, with convolutional and pooling layers, captures bone texture patterns for osteoporosis detection. Trained via backpropagation and gradient descent, it minimizes errors to accurately classify bone density disorders. Although this approach has demonstrated significant diagnostic potential, its limitation lies in the insufficient accuracy of BMD prediction, resulting in variability in sensitivity and specificity across different datasets. The variability can be attributed to factors such as differences in image quality, patient demographics, and the inherent limitations of using BMD as a single reference metric. Future work could focus on improving the model’s accuracy by incorporating additional imaging modalities, enhancing data preprocessing techniques, and exploring more advanced neural network architectures. Additionally, addressing the variability in sensitivity and specificity across datasets would require larger and more diverse training datasets to ensure robust generalization.

Wang et al. [19] developed a deep learning model with anatomical awareness; the model predicts the BMD by analyzing multiple bone-containing regions (ROIs), strategically selected to cover key anatomical structures relevant to the bone density. The Attentive Multi-ROI model utilizes deep learning, specifically the CNN, to extract and analyze features from chest X-ray images. The innovation of the Attentive Multi-ROI model lies in its anatomical awareness mechanism. By incorporating anatomical prior knowledge, the model can more accurately locate and analyze anatomical regions relevant to the BMD. By incorporating an attention mechanism, it automatically identifies and focuses on the most relevant regions, enhancing the prediction accuracy and reliability. However, despite the technical innovation of this model, its datasets were derived solely from healthy populations. This means that the image features and BMD distributions used for training and validation may differ from those encountered in clinical practice, especially in the presence of pathological conditions. This could limit the model’s predictive capability when applied to non-healthy populations and may fail to accurately reflect the actual BMD status.

Studies related to BMD detection have predominantly focused on the hip, whereas attention to the individual detection of single spinal vertebrae has been limited. For spinal BMD prediction and assessment, existing studies have primarily emphasized screening and diagnostic purposes rather than precise measurements, as the location of the spine and the presence of surrounding organs introduce significant complexity to the image background. This emphasis has resulted in a notable gap in the accurate prediction of single vertebral BMD.

Complex backgrounds in spinal X-ray images hinder accurate analysis and feature extraction, making it difficult to distinguish relevant features from noise. This interference affects the reliability of spinal BMD predictions by obscuring crucial bone details from ad-jacent organs and intricate backgrounds. To address these issues and enhance the feature ex-traction for vertebral BMD prediction, this study developed a hybrid model that integrates U-Net with artificial neural networks (ANNs). The U-Net model, which has demonstrated effectiveness in medical image segmentation tasks, was first employed for image preprocessing. It effectively reduces background noise and enhances bone structure features, thereby minimizing the influence of non-target regions on BMD prediction. Focus was placed on the individual detection and analysis of the L2 vertebra, ensuring targeted feature extraction from this region for enhanced precision. Subsequently, the ANN model processes the extracted features using deep learning techniques and applies nonlinear regression analysis to predict BMD values with greater accuracy. Unlike most existing studies that require both anteroposterior (AP) and lateral X-ray images to enhance the accuracy, this method successfully overcomes the challenges posed by spinal complexity by utilizing only AP-view X-ray images. This innovation simplifies the data acquisition process, reduces the dependency on multiple imaging views, and enhances the model’s practicality and generalizability. This combined approach aims to improve the precision of osteoporosis screening and assessment, with specific improvements in L2 vertebra analysis, allowing for a more precise assessment of early osteoporotic changes and providing stronger clinical support for timely intervention.

## 3. Materials

The dataset utilized in this study comprised 300 spine X-ray images, each with a pixel resolution of 256 × 256, extracted from the abdominal X-ray scans of various patients. These images were collected between January 2021 and July 2022 at the Hanyang University Medical Center, Seoul, Republic of Korea. Owing to the higher prevalence and increased risk of spinal conditions among females, the focus was placed on female patients. This study also showed that women are more susceptible to certain spine-related diseases, particularly osteoporosis, making this group critical for understanding and improving diagnostic accuracy [20].

Reference BMD data for the dataset were collected using a DXA scanner (HOLOGIC, Discovery W model), with BMD values recorded for four regions of the spine (L1, L2, L3, and L4) following a single DXA scan. This study ensured that the DXA scans and corresponding X-ray images were acquired simultaneously for the same patients. This pairing of data allowed for direct comparisons and more precise correlation analysis between DXA-derived BMD values and features extracted from X-ray images, facilitating the development of more accurate predictive models.

Additionally, variations in the BMD can be influenced by physiological factors, further affecting the measurement accuracy. Deng et al. [21] and Kim et al. [22] identified the bone area, body weight, and BMI as significant factors affecting the BMD. However, body weight was more easily recorded and exhibited a higher correlation with the BMD than BMI, which is why BMI was excluded from the dataset. Therefore, this study considered the bone area and body weight as the primary biological factors influencing the BMD. Table 1 summarizes the patients’ baseline characteristics according to sex.

## 4. Methods

In this study, an innovative approach was employed by integrating the U-Net and ANN models to accurately extract BMD values from spinal X-ray images. The primary objective was to address the challenges posed by the complex backgrounds of spinal X-ray images. To achieve this, we utilized the deep learning architecture of the U-Net model, leveraging multiple convolutional layers to extract key image features and employing skip connections to enhance detail capture. This unique network structure significantly improved the accuracy of feature identification. Each vertebral region of the spine image was subjected to both horizontal and vertical segmentation, encompassing the entire vertebral image. This segmentation process resulted in five distinct regions, each of which underwent a comprehensive analysis to extract detailed feature information. These features were then used as inputs for the ANN model, which employed its learning algorithms to comprehensively analyze the input features and produce accurate predictions of the BMD values for the L2 vertebra. Figure 1 displays the overall flowchart of our proposed method for measuring the BMD on X-ray images of the spine.

### 4.1. Extraction of Soft Tissue Using U-Net Model

X-ray imaging plays an indispensable role in medical diagnosis because of its ability to produce images with varying contrasts, which arise from the differences in X-ray absorption across different body tissues. This contrast is crucial for disease detection. DXA is widely regarded as the gold standard for BMD measurements. DXA utilizes the differential absorption of X-rays by the soft tissue and bone at two distinct energy levels, enabling highly accurate BMD calculations. By effectively isolating and removing the influence of soft tissues, DXA provides precise BMD values based on the principle that bone tissue absorbs more X-rays than soft tissue. However, this approach is not applicable to single-energy X-ray images, which lack the ability to differentiate or eliminate the effects of soft tissue interference, limiting its utility in direct BMD assessment.

In abdominal X-rays, the overlapping of various organs and the uneven distribution of background grayscale values present significant challenges for extracting the bone texture features essential for accurate BMD prediction. These complexities often obscure the subtle bone structures that are necessary for precise analysis. To overcome this limitation, advanced deep learning techniques were leveraged to effectively differentiate between bone and soft tissues. By minimizing the influence of soft tissue interference, our approach enhances the visibility of bone structures, leading to improved accuracy in L2 BMD prediction. The deep learning model not only refines the image contrast but also facilitates the automatic extraction of critical bone features, making it a robust tool in clinical imaging analysis.

To train the deep learning model, a synthetic dataset was constructed comprising the following two parts: input images (raw X-ray images of the spinal region) and label images (showing only soft tissue absorption). As it is impractical to obtain both components simultaneously at the same location, 150 abdominal X-ray images were first collected, focusing exclusively on the soft tissue, with particular emphasis on areas adjacent to the spine to accurately simulate the surrounding soft tissue environment. Concurrently, a total of 150 images of individual spinal partitions with uniform background grayscale values were acquired. Given the variability in bone density and morphological differences among the spinal partitions, four identical spinal partitioned images, each adjusted for varying degrees of contrast, brightness, and angular tilt, were used to create a composite mask representing the bone structures. To further enhance the dataset diversity, rotations and flips were applied to these processed bone templates, and they were combined with soft tissue images, resulting in a comprehensive synthetic dataset with diverse variations, as shown in Figure 2. This approach enabled the creation of a high-quality synthetic dataset that not only included essential image features, but also simulated real-world complexity, thereby providing a solid foundation for training deep learning models.

To minimize the influence of soft tissues on spinal X-ray images, this study employed the U-Net model, a convolutional neural network widely used for image partition. The U-Net model demonstrated significant benefits in our study because of its ability to leverage the advantages of traditional unsupervised learning networks in data dimensionality reduction while integrating both low- and high-level features for precise image processing. The architecture of U-Net is notably effective in addressing the limitations of traditional linear model-based correction methods and in handling complex nonlinear relationships with ease. This capability is particularly crucial for dealing with the complexity of soft tissue distribution, particularly in cases of tissue overlap and similar densities, where the performance of U-Net is exceptionally robust.

The U-Net model, with its meticulously designed encoder–decoder architecture, is adept at extracting and reconstructing image features with a high precision. The encoder part of the network uses a series of convolutional and pooling layers to progressively extract higher-level features while reducing the spatial resolution. Each convolutional layer involves two 3 × 3 convolutions with rectified linear unit (ReLU) activation, followed by 2 × 2 max pooling. This process enables the network to downsample the feature maps and capture increasingly abstract representations. In contrast, the decoder uses upsampling techniques to restore the image to its original dimensions. It incorporates skip connections to transfer feature maps directly from the encoder to the corresponding layers in the decoder, thereby preserving and utilizing detailed spatial information. The output layer of the U-Net model employed 1 × 1 convolutions to map the feature maps to the pixel classification results for the target classes. A stochastic gradient descent (SGD) algorithm was employed to optimize the U-Net model. SGD updates the model parameters in an iterative manner to minimize the loss function, as shown in Equation (1), where represents the parameters at iteration is the learning rate, and is the gradient of the loss function with respect to the parameters.(1)$$\theta_{t+1}=\theta_{t}-\eta {◸}_{\;\theta }J\left(\theta_{t}\right)$$

To quantify the performance of the model, the mean squared error (MSE) served as the loss function of choice to measure the difference the between predicted and actual images by applying a greater penalty to larger errors, which improved the accuracy of the model in generating soft tissue images. The MSE gauges the average squared discrepancy between the predicted and actual values, as expressed in Equation (2).(2)$$MSE=\frac{1}{N}\sum_{i=1}^{N}{\left(y_{i}-{\hat{y}}_{i}\right)}^{2}$$
where is the total number of samples, is the actual value, and is the predicted value of the model. The MSE metric serves as a critical indicator of the accuracy of the model, guiding the optimization process towards minimizing the prediction error.

After multiple training epochs, the U-Net model uses spinal X-ray images as the input and generates soft tissue images as the output, allowing the bone tissue-dominant X-ray images to be obtained by subtracting the generated soft tissue images from the original spinal images.

### 4.2. ANN Model Utilization for BMD Prediction

For an in-depth analysis of the microscopic features of the spine, image partition techniques were employed to reveal texture variations across different regions. To capture richer information on the texture characteristics of the vertebral body, precise regional partitioning was implemented. To minimize the background interference, the analysis focused on the central region of the spine. Hulme et al. [23] indicated that trabecular changes primarily occur in the central and upper parts, leading to the division of the spine into the following five regions: upper, lower, left, central, and right. The mean grayscale value of each region was calculated and used as input features for the ANN model, with the goal of improving the feature extraction accuracy and enhancing the scientific basis for clinical diagnosis.

During the spine image partitioning process, thresholding techniques were employed to enhance the visualization of spinal structures and extract key information. Thresholding is an efficient image binarization method that separates the image into the foreground and background. This method is particularly important in spinal image analysis, as it not only highlights the texture features of the spine but also effectively suppresses background noise, thereby significantly improving the accuracy of subsequent image processing and analysis.

Drawing from similar methods in Nguyen et al. [24], three thresholds were set to process the images of the five partitioned regions. The thresholds were selected based on the distribution of grayscale values within the segmented spinal regions, enabling the determination of an optimal threshold range. To ensure the preservation of image details, three threshold ranges for image processing were defined, derived from the standard deviation of the Gaussian function, as follows:$$Threshold\;Set\;1:\;\left[\mu \left(G\right)-std\left(G\right),\mu \left(G\right)+1.96\times \sigma \left(G\right)\right],\;Threshold\;Set\;2:\;[\mu (G)-1.96\times \sigma (G),\mu (G)+1.96\times \sigma (G)],\;Threshold\;Set\;3:\;[\mu (G)-1.96\times \sigma (G),\mu (G)+\sigma (G)]$$
where and represent the mean and standard deviation of the grayscale values, respectively. In the processed regions, the grayscale values at pixel coordinates must satisfy the following conditions to fall within the defined threshold range, as shown in Equation (3):(3)$$G(i,j)=\left\{\begin{array}{c} A1,\;if\;G(i,j)<A1\\ A2,\;if\;G(i,j)>A2\\ G(i,j),\;otherwise\; \end{array}\right.$$

Here, denotes the grayscale value at pixel, and the adjustments are applied to limit the values to the threshold range [A1, A2]. After applying the threshold, the average grayscale value of each region’s image is extracted and used as input features for the ANN model.

To construct the predictive model, the weight and vertebral area were selected as key input features due to their strong correlation with the BMD, as identified in previous studies by Deng et al. [21] and Kim et al. [22]. These factors were prioritized to maintain model efficiency while minimizing the complexity, as incorporating additional low-correlation features could introduce noise, hinder training, and reduce the generalization performance. Additionally, to enhance the feature extraction accuracy, image partitioning techniques were employed to analyze regional variations in the vertebral texture. Hulme et al. [23] highlighted that trabecular changes predominantly occur in the central and upper parts of the spine, guiding the division of vertebral images into the following five regions: upper, lower, left, central, and right. The mean grayscale value of each region, which serves as a key indicator of the BMD distribution across different areas, was computed and used as an additional input feature for the ANN model. By focusing on the central region of the spine, this approach minimized background interference and ensured that the extracted features effectively captured the variations in bone mineral density, thereby strengthening the scientific foundation for clinical diagnosis.

The ANN comprises multiple layers of neurons, with each layer connected by weighted links that transmit feature information from spinal images to subsequent layers. In our ANN model, the ReLU activation function is employed in the hidden layers to introduce nonlinearity. The ReLU function is expressed in Equation (4) as follows:(4)f(x)=max(0,x)

This activation function helps to mitigate the vanishing gradient problem by allowing only positive inputs to pass through, enhancing the training efficiency of deep networks. In the output layer of the ANN model, a modified version of the standard tanh activation function, is used to constrain the predicted values within a specific range. The custom activation function is formally defined in Equation (5). Additionally, the hyperbolic tangent (tanh) function, commonly used for its smooth gradient and bounded output, is given by Equation (6), as follows:(5)$$f\left(x\right)=0.55\cdot \tanh\left(x\right)+0.85$$(6)$$\tanh\left(x\right)=\frac{e^{x}-e^{-x}}{e^{x}+e^{-x}}$$

The constant 0.55 controls the amplitude of the function in Equation (6), ensuring that the output range is compressed to approximately 1.1 units. The constant 0.85 shifts the function upward, ensuring that the final output values lie within the desired range of 0.3 to 1.4, for constraining the predicted values to remain within a biologically realistic range. The ANN model consists of an input layer with 23 neurons, encompassing all extracted image features and patient physiological data. It includes three hidden layers designed to capture complex feature interactions. The output layer employs a modified tanh activation function for enhanced prediction stability and accuracy. For further details, refer to Figure 3.

During the performance evaluation, the dataset was meticulously divided into training and validation sets at a 1:4 ratio, ensuring robust generalization to unseen data. The BMD values from the DXA scans were used as target variables for training and validating the ANN model. To prevent overfitting, we applied early stopping, which monitors the validation loss and halts training if no significant improvement is observed over consecutive epochs [25]. This not only enhances the generalization but also improves the model stability and efficiency. Throughout the training process, the model’s loss function exhibited a consistent downward trend, indicating effective learning, while the validation loss remained stable with minimal fluctuations, further affirming the model’s ability to generalize well to new data without significant overfitting. As illustrated in Figure 4, these results visually demonstrate the model’s robustness and the effectiveness of its learning process.

## 5. Result

### 5.1. Result of U-Net Model Application

By extracting image features from individual vertebrae across various vertebrae and incorporating relevant physiological factors, the dataset was partitioned into training and validation sets for input into an ANN model. This process enabled the generation of predicted BMD values for the L2 vertebra. To evaluate the linear relationship between the predicted and actual BMD values, both before and after applying the U-Net model, Pearson’s correlation coefficient was employed. This coefficient measures the degree of linear similarity between predicted and observed values, ranging from −1 to 1, where 1 indicates a perfect positive correlation and 0 signifies no correlation.

Through in-depth model analysis and prediction of the L2 vertebra, we demonstrated its outstanding performance in BMD prediction. Specifically, for the L2 vertebra, the Pearson correlation coefficient on the validation set reached 0.770 and the mean absolute error (MAE) was 0.08, as shown in Figure 5. These metrics not only represent the model’s optimal performance in accurately predicting the BMD for the L2 vertebra but also highlight its robustness in handling complex biomedical data. To clearly demonstrate the contribution of the U-Net model in mitigating the effects of soft tissue, a detailed comparison of the results for the L2 vertebra was performed both before and after applying the U-Net model. The original images, prior to the application of the U-Net model, were also used to predict the BMD results, enabling a comparison that more effectively demonstrated the performance of the U-Net model. The results using the original images yielded a Pearson correlation coefficient of 0.441 for the validation set, and the MAE was 0.084, as shown in Figure 5.

The overall prediction accuracy was greatly improved due to better feature isolation and the reduction in soft tissue interference. These results validate the model’s effectiveness in improving bone density measurement performance for the L2 vertebra. The L2 model demonstrates outstanding performance in BMD prediction, particularly in handling complex backgrounds and noise, and in preserving essential spatial details. These findings offer valuable insights for future research and clinical applications, providing new perspectives and methods for bone density measurement and related disease diagnosis.

### 5.2. Comparison of Results Across Different Vertebra

To more comprehensively demonstrate the model’s performance, in-depth analysis and prediction were conducted for other vertebrae as well. This comparison aimed to evaluate the model’s effectiveness in improving the BMD prediction accuracy by examining the variations across different vertebrae. By analyzing the changes in the prediction performance, we gained a better understanding of how the U-Net model enhances the bone density measurement accuracy and addresses the challenges related to soft tissue interference. The following sections present the results of applying the model to different vertebrae, further highlighting its performance across various regions of the spine. Specifically, the L1 model achieved a coefficient of determination (R^2^) of 0.521. For L3, it achieved an R^2^ of 0.602, while, for L4, it achieved an R^2^ of 0. 485, as shown in Figure 6. When examining the MAE values across the vertebrae, L2 showed an increase from 0.08 to 0.084. Other vertebrae also exhibited significant changes. For L1, the MAE rose from 0.076 to 0.084; for L3, it slightly decreased from 0.099 to 0.095; for L4, it increased from 0.116 to 0.122, as shown in Figure 7.

The model demonstrated relatively good predictive results, indicating its effectiveness in retaining critical skeletal structure information from the vertebrae and performing accurate analysis. Among the three vertebrae, the best performance was observed for L3. However, the performance for all vertebrae did not surpass that of the L2 vertebra. This highlights the necessity of conducting analysis and prediction for individual vertebrae, emphasizing the importance of focusing on specific vertebral regions for more accurate and precise BMD estimation.

### 5.3. Comparison of Results Between Autoencoder and U-Net

In the study by Nguyen et al. [24], an Autoencoder was applied for the BMD prediction in the hip region, with the aim of simplifying the complexity of the input images through tasks like denoising and dimensionality reduction. Although the Autoencoder performed well in these tasks, its performance was less satisfactory when dealing with spinal X-rays due to the complexity of the background and anatomical structures. In particular, the Autoencoder struggled to retain the spatial details that are critical for accurate BMD prediction. Our detailed analysis of the BMD prediction, as shown in Figure 8, focuses on the L2 vertebra. After applying the Autoencoder model, the Pearson correlation coefficient (R^2^) for L2 decreased significantly from 0.77 to 0.534, indicating a substantial decline in the predictive accuracy, as shown in Figure 8. This reduction in R^2^ suggests that the Autoencoder model was less effective in capturing the relationship between the predicted and actual BMD values compared to the U-Net model.

In contrast, when comparing the MAE values across different vertebrae, L2 exhibited the most pronounced increase, rising from 0.08 to 0.102. The changes in MAE values for the other vertebrae were also notable. For L1, the MAE increased from 0.076 to 0.084; for L3, it rose slightly from 0.094 to 0.095; for L4, it increased from 0.116 to 0.133, as shown in Figure 8. These results indicate that, compared to the U-Net model, the Autoencoder led to higher MAE values across all vertebrae, further highlighting the decline in the predictive performance, particularly for L2. It is hypothesized that the key reason for this discrepancy lies in the additional mechanism provided by the U-Net model, specifically the use of skip connections. These skip connections serve as a bridge between the encoder and decoder stages, allowing U-Net to preserve and transmit essential spatial information more effectively. This feature significantly enhances the model’s ability to distinguish bone from surrounding soft tissues, particularly in areas with complex anatomical structures and background noise.

To provide a clearer demonstration of the performance differences between these two models, a comparative analysis of their results in predicting BMD values was conducted. This analysis highlighted how U-Net’s architectural advantages, especially its ability to retain spatial detail, lead to improved accuracy in BMD prediction, particularly for spinal X-ray images.

## 6. Discussion

Monitoring the BMD as a common method of assessing conditions, such as osteoporosis, is widely recognized. In this study, we adopted an innovative approach using a U-Net model to process spinal X-ray images coupled with thresholding techniques to extract multi-level image features specific to the L2 vertebra. Additionally, patient weight and the area of vertebra were considered as key input features. These combined features were used to train a dedicated ANN model focused exclusively on the L2 vertebra. This targeted approach accounted for the unique structural and environmental characteristics of the vertebra, ensuring data consistency and enhancing both the prediction accuracy and reliability. Our model was implemented on a computer equipped with an Intel i7-12700 processor, an NVIDIA RTX 3070 GPU, and 32 GB of RAM. Leveraging GPU acceleration, the model achieved efficient training and inference, ultimately reaching a prediction time of 0.003349 s per sample. This highlights its computational efficiency and suitability for real-time clinical applications. By focusing on the single vertebra, the model effectively captured subtle details crucial for improving the precision of BMD predictions, thereby facilitating the early diagnosis and treatment of osteoporosis, and contributing to the prevention of spinal fractures and their associated complications.

With the integration of deep learning algorithms in the field of artificial intelligence, spinal X-ray analysis has emerged as a mainstream approach for accurately diagnosing and monitoring various diseases. This innovative application offers significant advantages, particularly in overcoming limitations, such as high equipment costs and limited accessibility. However, compared with regression-based methods for predicting BMD values, most X-ray image analyses focus on classifying healthy and diseased states. Although they demonstrate good performance in terms of precision and accuracy, concerns remain regarding their reliability in clinical practice, as several models struggle to achieve a high predictive accuracy in real-world settings. For preventing osteoporosis-related complications, most classification models are limited to detecting advanced disease stages, focusing primarily on distinguishing between healthy and diseased states [26,27,28]. Although these models perform well in terms of accuracy for severe cases, they often fail to provide early detection or identify subtle changes in bone density that occur in the early stages of the disease. This limitation hinders the development of proactive treatment and prevention plans, as early intervention is critical for mitigating the long-term impact of osteoporosis. To address this gap, predicting the BMD values of individual spinal vertebrae offers a more effective and targeted approach for assessing fracture risk, enabling personalized care tailored to patient-specific conditions. By focusing on individual vertebrae, this method captures localized changes in bone density that might be overlooked in broader classification models, allowing for the earlier detection of potential fracture sites and the development of more precise prevention and treatment strategies.

In the exploration of deep learning techniques for detecting the BMD in spinal images, the study conducted by Nguyen et al. [24] successfully employed a convolutional Autoencoder model to enhance the images, leading to a substantial improvement in the image quality. By extracting features from the enhanced images, the researchers were able to predict the BMD values with a high degree of accuracy. The study reported a correlation coefficient of 0.81 between the predicted and actual BMD values, highlighting the model’s strong potential in predictive performance. Notably, the study primarily focused on image processing in the hip region, where the grayscale distribution is more uniform compared to the spine. This uniformity facilitates the extraction of detailed information, making image analysis in the hip region clearer and more precise.

Zhou et al. [29] proposed a hybrid deep learning framework (HDLF) for BMD prediction and classification using biplanar X-ray (BPX) images. By integrating CNN-based spatial feature extraction with clinical data, the model effectively captured osteoporosis-related risk factors. Trained on 906 BPX scans from 453 subjects, with QCT as the reference standard, the HDLF achieved R^2^ values of 0.77 and 0.79 for BPX-only and multimodal predictions, respectively, with Pearson correlation coefficients of 0.88 and 0.89. The classification model demonstrated an AUC of 0.97, an accuracy of 0.93, and an F1 score of 0.93, highlighting its potential as an automated osteoporosis screening tool. Additionally, the study evaluated BMD prediction using only AP-view X-ray images, yielding an R^2^ value of 0.69, indicating a relatively lower correlation. In comparison, our study focused on spinal BMD prediction from abdominal X-rays, which are more commonly available in clinical practice than lateral or anteroposterior spinal X-rays. However, these images pose greater challenges due to complex backgrounds and noise. To address this, we incorporated U-Net for preprocessing, effectively reducing soft tissue interference and improving the prediction accuracy, demonstrating the feasibility of utilizing abdominal X-rays for BMD assessment.

Peng et al. [30] advanced osteoporosis diagnosis using the VB-Net structure for the automatic segmentation of spinal CT images, achieving a high BMD prediction accuracy, with R^2^ values of 0.991, 0.962, and 0.878 for the training set, test set, and overall performance, respectively. While the superior quality of CT images enhances the prediction accuracy, their high cost and limited accessibility restrict the clinical applicability. Additionally, their study focused on predicting the overall BMD rather than analyzing and predicting the BMD for individual vertebrae. In contrast, our model utilized X-ray images, which are more accessible in clinical practice, and employed a global analysis approach to preserve the structural information, thereby improving the prediction accuracy. Although our correlation coefficient is slightly lower, our model demonstrates strong generalization across datasets, providing a more cost-effective and efficient solution for clinical application. To further illustrate the performance of our model, we present a comparative analysis with the above method, as shown in Table 2.

This body of work highlights the ongoing advancements in applying deep learning to spinal imaging for osteoporosis diagnosis, while also addressing the challenges and limitations of image quality, segmentation strategies, and the inherent complexity of spinal structures. Our study has made significant advancements in the accuracy of BMD measurements, offering an innovative tool for the early diagnosis and treatment of osteoporosis, which holds immense importance in the current medical research field. By employing this advanced method, clinicians can provide more precise diagnostic information, enabling the development of more effective treatment plans. However, our study also has certain limitations. Due to the higher prevalence of osteoporosis in women, female cases were more readily available during data collection, leading to a higher proportion of female samples in our dataset. This imbalance may affect the generalizability of our findings, highlighting the need for more diverse data in future research. However, the predictive correlation for other vertebrae was relatively low, indicating room for improvement. There were notable challenges in this study that could have influenced the errors in the results. First, eliminating the influence of soft tissues on imaging remains a complex issue. The clinical evaluation of abdominal X-ray images becomes more complicated owing to variations in the density and shape of the abdominal organs [31]. The comparison of the U-Net model’s segmentation performance on spinal X-ray images with simple and complex backgrounds highlights the impact of background complexity on segmentation results, as shown in Figure 9. The comparison of the U-Net model’s performance on spinal X-ray images with simple and complex backgrounds, as shown in Figure 9, highlights the impact of background complexity. Figure 9 illustrates the original images and their corresponding segmentation results, showing effective extraction for simple backgrounds and reduced performance for complex backgrounds. Good extraction is observed in the simple background, where the clear contrast between bone and tissue allows for accurate segmentation, while poor extraction is seen in the complex background, where non-bone tissue and brightness variation complicate the process. Moreover, the structural overlap and small signal-to-noise ratios in abdominal radiographs necessitate the use of specialized skeletal X-ray techniques to obtain clearer images [32,33]. As individuals age, muscle strength gradually weakens, and spinal curvature may occur, introducing additional complexity to abdominal X-ray images in elderly datasets. Increased organ overlap and changes in abdominal contours owing to muscle weakness can result in an uneven grayscale distribution in images, further complicating image processing.

Additionally, the influence of bone diseases on BMD values cannot be overlooked. Bone diseases, particularly osteophyte formation, can lead to abnormal variations in bone density. Osteophytes are additional bone growths that form at the edges of bones and are often associated with degenerative changes in vertebra or osteoarthritis. Studies have shown that DXA equipment may mistake osteophytes for part of the vertebrae during measurements, thereby affecting the accuracy of the BMD estimation and prediction. Among the four lumbar vertebrae (L1–L4), L4 had the highest prevalence of osteophytes across all age groups, indicating a significant impact on the DXA-measured BMD values [32,34]. Furthermore, the presence of osteophytes across different vertebrae can interfere with the accuracy of BMD measurements [35,36].

Additionally, the choice of the spinal X-ray imaging position may have a negative impact on the results. Studies have shown that the anterior portion of the vertebral body is primarily composed of trabecular bone, whereas the posterior portion mainly consists of cortical bone, resulting in a faster rate of bone loss in the anterior section than in the posterior section [37]. Therefore, when assessing age-related bone loss, lateral BMD measurements using DXA have demonstrated greater sensitivity than traditional anteroposterior (AP) scans [38]. This discrepancy can lead to changes in spinal morphology, particularly under mechanical stress. The variation in bone loss rates poses challenges for the accurate detection of BMD values using AP X-ray images.

## 7. Conclusions

In this study, a U-Net-based image segmentation technique was successfully developed that was capable of efficiently isolating muscle and fat tissues from X-ray images. This significantly improved the precision of the bone region extraction and effectively reduced the interference of non-target areas on the BMD predictions. Additionally, by combining bone image features with key physiological parameters, such as patient weight and individual vertebrae area, and inputting them into the ANN for nonlinear regression analysis, we achieved a substantial improvement in the BMD prediction accuracy and enhanced the model’s generalizability across diverse patient groups and datasets.

Consequently, our automated measurement technique for diagnostic applications has successfully introduced the following key innovations and capabilities:U-Net enhancement technique: The U-Net-based image enhancement technique developed in this study significantly improves the feature recognition accuracy by precisely extracting and reconstructing image features. This technique leverages the strengths of the U-Net model’s encoder–decoder architecture, employing multiple convolutional and pooling layers to deeply explore high-level image features. Additionally, through skip connections, the model effectively enhances the capture of detailed information.Nonlinear regression analysis with the ANN: In this study, bone features extracted from images were combined with key physiological parameters, such as patient weight and the individual vertebra area, and input into the ANN for nonlinear regression analysis. By introducing nonlinearity through ReLU activation functions in the hidden layers and applying a customized version of the tanh activation function in the output layer to constrain the predicted values, we not only improved the model’s predictive stability but also significantly enhanced the accuracy of the predictions.

## Figures and Tables

**Figure 1 bioengineering-12-00385-f001:**
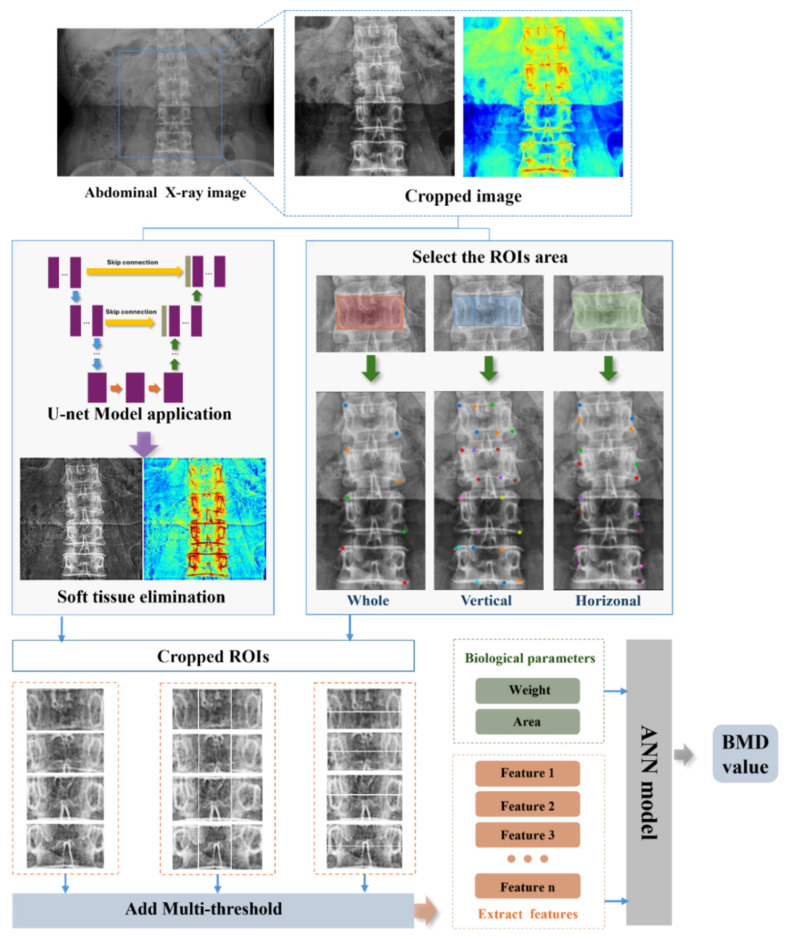
Overall flowchart of our proposed method for measuring the BMD in X-ray images of the spine.

**Figure 2 bioengineering-12-00385-f002:**
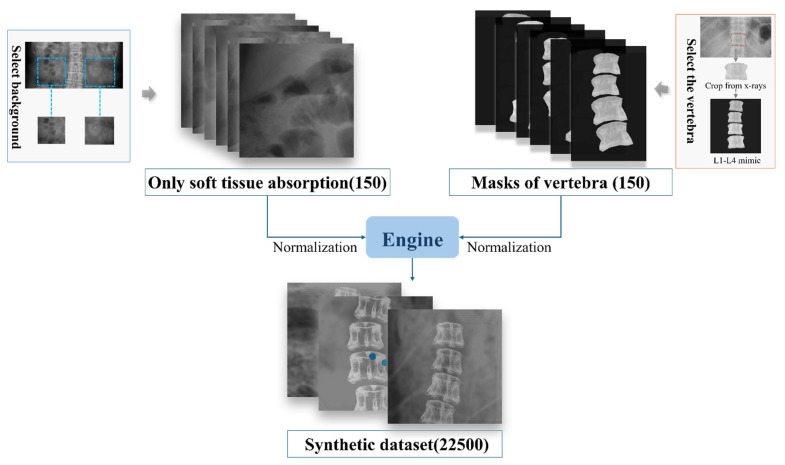
Procedure for generating synthesis images for training.

**Figure 3 bioengineering-12-00385-f003:**
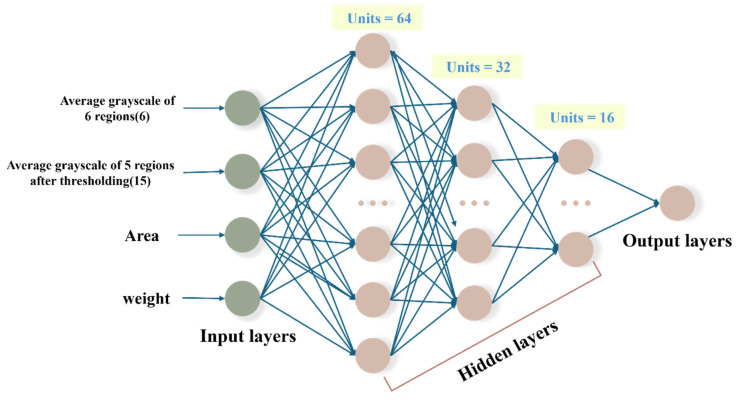
The structure of the ANN model.

**Figure 4 bioengineering-12-00385-f004:**
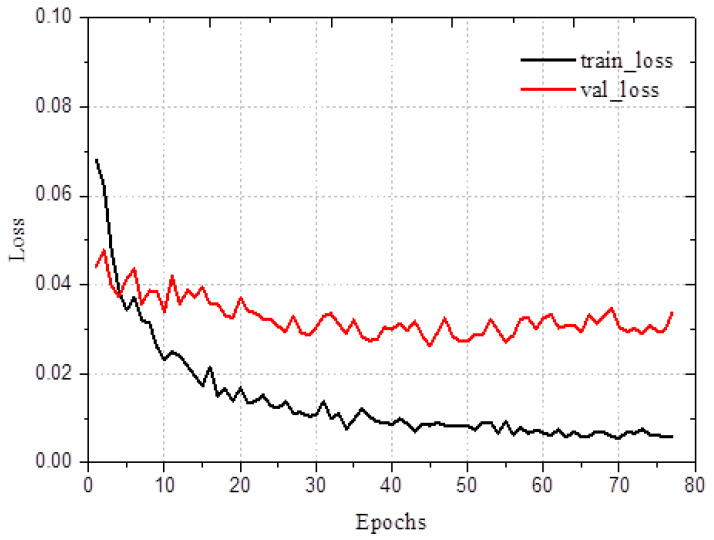
Training and validation loss trends in the ANN model.

**Figure 5 bioengineering-12-00385-f005:**
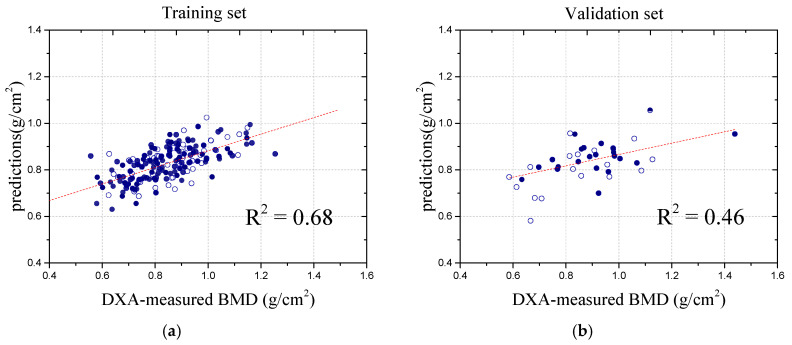
Comparison of correlation coefficients of L2 before and after U-Net application. (**a**) Training result before using the model. (**b**) Validation result before using the model. (**c**) Training result after using the model. (**d**) Validation result after using the model.

**Figure 6 bioengineering-12-00385-f006:**
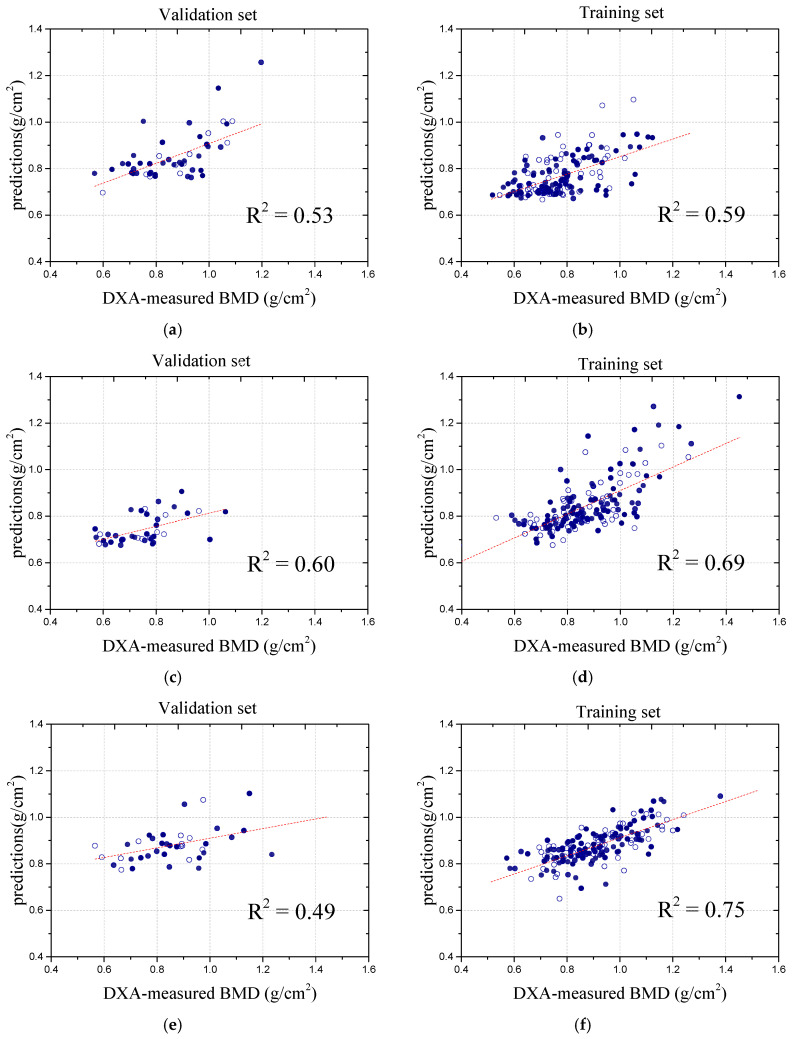
Comparison of correlation coefficients across different vertebra after applying the U-Net model. (**a**) Validation result of L1; (**b**) Training result of L1; (**c**) Validation result of L3; (**d**) Training result of L3; (**e**) Validation result of L4; (**f**) Training result of L4.

**Figure 7 bioengineering-12-00385-f007:**
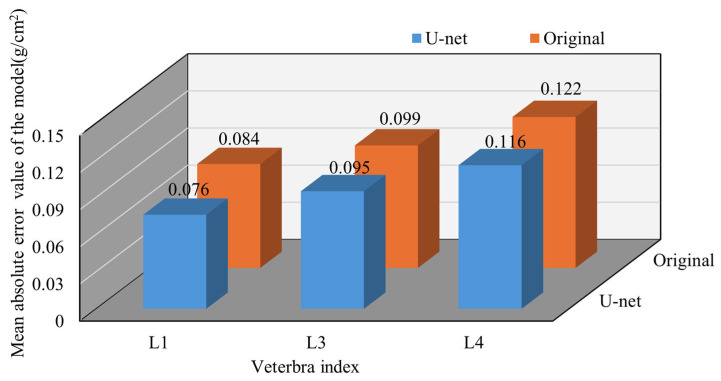
Comparison of MAE values for each vertebra before and after using the U-Net model.

**Figure 8 bioengineering-12-00385-f008:**
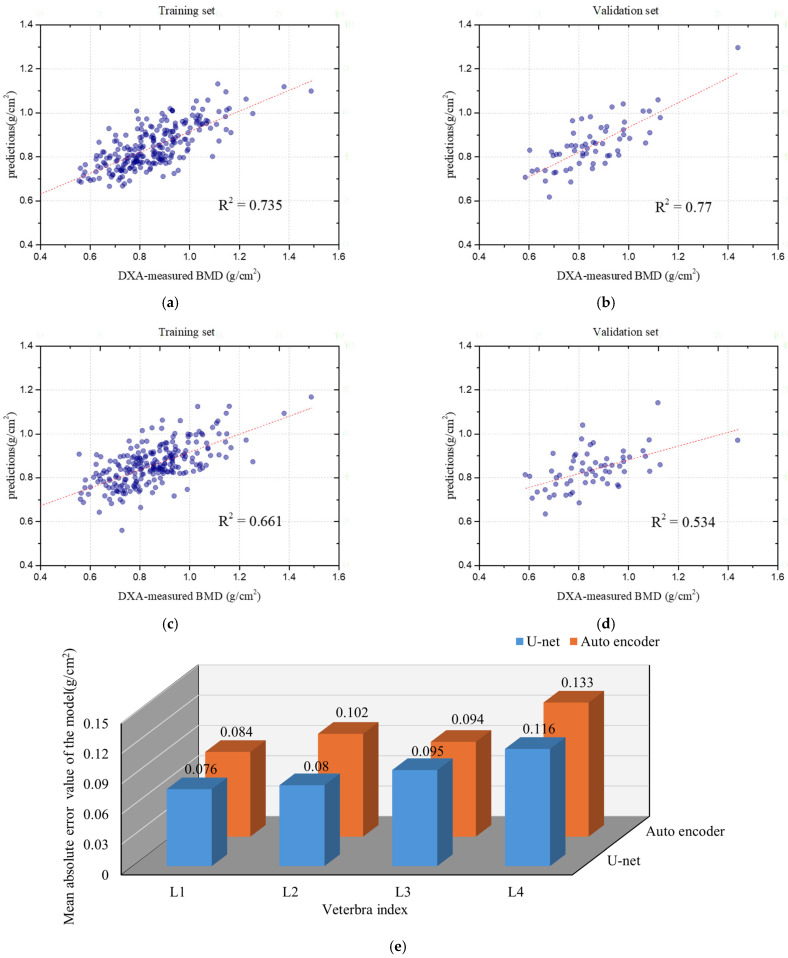
Comparison of correlation coefficients of L2 and the MAE values for each vertebra after using the U-Net model and using the Autoencoder: (**a**) Training result of using U-Net. (**b**) Validation result of using U-Net. (**c**) Training result of using the Autoencoder. (**d**) Validation result of using the Autoencoder. (**e**) Comparison of MAE values.

**Figure 9 bioengineering-12-00385-f009:**
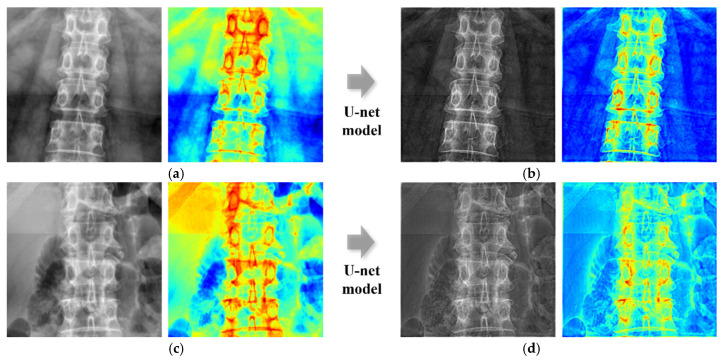
Comparison of good and poor bone extraction effects in the U-Net model. (**a**) Original X-ray image (good extraction). (**b**) U-Net segmentation result (good extraction). (**c**) Original. X-ray image (poor extraction). (**d**) U-Net segmentation result (poor extraction).

**Table 1 bioengineering-12-00385-t001:** Statistical data for participating patients and dataset distribution.

Parameters		Patient Gender	
		Female	Male
Number of cases		275	25
Average body weight ± STD (kg)		55.66 ± 8.89	62.6 ± 13.46
Average bone area ± STD (cm^2^)	L1	12.58 ± 1.47	15.18 ± 1.52
	L2	13.57 ± 1.37	16.14 ± 1.99
	L3	14.83 ± 1.54	17.08 ± 1.82
	L4	16.24 ± 1.85	18.93 ± 2.37
Dataset distribution	Total	300	
	Training set	240 (80%)	
	Validation set	60 (20%)	

**Table 2 bioengineering-12-00385-t002:** Comparison of other previous methods for BMD prediction.

Study	Model	Data Type	Dataset Size	Performance Metrics	
				Correlation Coefficient (R^2^)	MAE (g/cm^2^)
Our study	U-Net + ANN hybrid model	Abdominal X-ray images	300 images	0.77 (L2), 0.521 (L1), 0.602 (L3), 0.485 (L4)	0.08 (L2), 0.084 (L1), 0.095 (L3), 0.122 (L4)
Nguyen et al. [24]	Convolutional Autoencoder	Hip X-ray images	673 images	0.7	0.083
Zhou et al. [29]	HDLF	BPX clinical data	906 BPX scans	0.69 (AP-only), 0.77 (BPX-only), 0.79 (Multimodal)	18.29(AP-only), 16.13 (BPX-only), 15.51(Multimodal)
Peng et al. [30]	VB-Net	Spinal CT images	1219 CT images	0.962	10.257

## Data Availability

No new data were created or analyzed in this study. Data sharing is not applicable to this article.

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
