# Peer review of "U-Net-Based Deep Learning Hybrid Model: Research and Evaluation for Precise Prediction of Spinal Bone Density on Abdominal Radiographs"

_bioengineering, 2025, doi:10.3390/bioengineering12040385_

Round 1

Reviewer 1 Report

Comments and Suggestions for Authors
  1. In the abstract mention the specific U-Net variation used. Have you come up with own U-Net variant?
  2. Abstract – indicate the significance of the proposed method, with the highest performance metric value obtained.
  3. You have stated that correlation coefficient is 77%. Is it sufficient for a medical related highly dependable system?
  4. The use of (citing) references has done consequently, adding references for each of the fact. It seems that you have not critically analyzed the literature well. Because, a given reference can be cited in many places. And a given fact can be referred by many references.
  5. Introduction: clearly state the research questions addressed in this work, highlighting the novelty of this study, compared to the latest studies.
  6. It would be better to include a new section2, to explain the related studies and theories behind them.
  7. In table 1, you have stated the details of the participants. But, it would be better to include the details of the considered dataset. The number of images in each class label. And the number of images in the training set, testing set. Discuss the data imbalance issues and how you addressed those.
  8. With Figure 4, how do you discuss the data overfitting/ underfitting issues. What precautions you have taken to avoid those issues.
  9. In your methodology, have you used the Original U-Net as it is? Are there any research contributions you have introduced.
  10. How do you justify the use of this U-net model, compared to other segmentation models with attention and feedback mechanisms.
  11. Discuss the computational complexity of this model.
  12. What is the practical feasibility of deploying this model in real world settings
  13. Compare the proposed model with existing studies, in the discussion section.
  14. Include more latest references, from 2025, 2024, 2023
Comments on the Quality of English Language

Proofread the paper

Reviewer 2 Report

Comments and Suggestions for Authors

1) lacks a detailed comparison of the proposed U-Net + ANN model against other state-of-the-art models in the field of bone mineral density (BMD) prediction. Strengthen the validity by a comparative table among the metrics such as accuracy, sensitivity, and specificity with other deep-learning models (e.g., CNN-based, Transformer-based)
2) this work based on female patients that may introduce gender bias in model predictions. As the osteoporosis is more common in women, authors are advised to discuss on how the model generalizes to male patients. Can you explainn about the potential gender-based differences in spinal BMD prediction?
3) In this manuscript, the artificial neural network structure's and feature selection process need more justification. for understanding , the choice of input features (like grayscale values, weight, vertebral area) should be supported with statistical analysis or feature importance ranking. Authors are advised to explain why those features selected over others (e.g., age, height, BMI)
4) Some figures are incorrectly cited (e.g., Figure 6 appears before Figure 5). Address the issues.

Round 2

Reviewer 1 Report

Comments and Suggestions for Authors

Paper is improved

Comments on the Quality of English Language

Proofread for language improvements

Reviewer 2 Report

Comments and Suggestions for Authors

Authors revised the manuscript as per suggestions